# Isolation, Identification, and Characterization of *Aeromonas veronii* from Chinese Soft-Shelled Turtle (*Trionyx sinensis*)

**DOI:** 10.3390/microorganisms11051304

**Published:** 2023-05-16

**Authors:** Xiaowei Hu, Zidong Xiao, Bo Li, Mingyang Xue, Nan Jiang, Yuding Fan, Peng Chen, Feng Qi, Xianghui Kong, Yong Zhou

**Affiliations:** 1Department of Fisheries Development, College of Fisheries, Henan Normal University, Xinxiang 453000, China; 2Yangtze River Fisheries Research Institute, Chinese Academy of Fishery Sciences, Wuhan 430223, China; 3Wuhan Academy of Agricultural Sciences, Wuhan 430071, China; 4Institute of Fishery Research of Xinjiang Uygur Autonenous Region, Urumqi 830099, China

**Keywords:** Chinese soft-shelled turtle, *Aeromonas veronii*, isolation, identification, histopathological analysis

## Abstract

*Aeromonas veronii* is widespread in aquatic environments and is capable of infecting various aquatic organisms. *A. veronii* infection is lethal for Chinese soft-shelled turtles (*Trionyx sinensis*, CSST). We isolated a gram-negative bacterium from the liver of diseased CSSTs, which was named XC-1908. This isolate was identified as *A. veronii* based on its morphological and biochemical characteristics, and 16S rRNA gene sequence analysis. *A. veronii* was pathogenic for CSSTs with an LD_50_ of 4.17 × 10^5^ CFU/g. The symptoms of CSSTs artificially infected with isolate XC-1908 were consistent with those of the naturally infected CSSTs. The levels of total protein, albumin, and white globule in the serum samples of the diseased turtles were decreased, whereas those of aspartate aminotransferase, alanine aminotransferase, and alkaline phosphatase were elevated. Moreover, the diseased CSSTs exhibited the following histopathological changes: the liver contained numerous melanomacrophage centers, renal glomerulus were edematous, intestinal villi were shed, and in oocytes, the number of vacuoles increased and red-rounded particles were observed. Antibiotic sensitivity tests revealed that the bacterium was sensitive to ceftriaxone, doxycycline, florfenicol, cefradine, and gentamicin, and resistant to sulfanilamide, carbenicillin, benzathine, clindamycin, erythromycin, and streptomycin. This study provides control strategies to prevent outbreaks of *A. veronii* infection in CSSTs.

## 1. Introduction

The Chinese soft-shelled turtle (CSST; *Trionyx sinensis*) belongs to the *Trionychidae* family [1], and more than 30 species have been reported for the genus *Trionyx*. However, only three species of CSST are found in China, namely *Peleochelys bibroni*, *Palea steindachneri*, and *T. sinensis* [2]. The CSST is one of the most significant aquatic species in China, and it is widely farmed for food production, with a production of 319,081 tons in 2018 [2]. Wild CSST species are mainly distributed in Asia, including China, Japan, eastern Russia, and Korea. However, with advancements in turtle farming, various turtle diseases have been reported, such as red bottom plate disease, putrid skin disease, etc. [3]. In a high-density intensive farming model with high temperature, the mortality rates of infectious turtle diseases are increasing, incurring huge economic losses. Every year during summer, turtle disease outbreaks, especially bacterial diseases [4] caused by *Aeromonas hydrophila* [5], *Bacillus cereus* [6], and *Edwardsiella tarda.* [7], affect the turtle farming industry.

The bacterium *Aeromonas veronii*, which belongs to the *Aeromonadaceae* family, was first isolated in 1983 from patients with diarrhea and wound infections by the Centers for Disease Control and Prevention (CDC) [8]. It was named *A. veronii* in the honor of the French microbiologist Veron [8]. The bacterium can adapt to different types of environments and can infect mammals including humans, especially the elderly, children, and individuals with low immunity [9]. Moreover, it can infect fish, amphibians, and reptiles, causing huge economic losses to the aquaculture industry [10,11]. *A. veronii* is widely distributed in the aquatic environment. Moreover, it multiplies faster under the following conditions: during summer and autumn, in water with deteriorated quality, and in physically traumatized or immunocompromised aquatic animals [12]. *A. veronii* infections exhibit high mortality rates, with hemorrhage and ascites being the main pathological features of the affected aquatic animals [13].

In 2022, at a farm in Hubei Province, CSSTs exhibited the symptoms of an infectious disease, such as floating on the surface, slow movement, easily catchable, and white spots on the abdominal surface. Therefore, in the present study, we determined the causal organism of the infection observed in CSSTs at the farm, under aseptic conditions. We identified the bacterium based on morphological, physiological, and biochemical criteria and 16S rRNA gene sequencing. Moreover, the pathogenicity and drug sensitivity of the bacterium were determined using the regression infection test and the agar diffusion method, respectively. The result of this study provides a basis for the development of control strategies for CSST diseases.

## 2. Materials and Methods

### 2.1. Trionyx Sinensis

Twelve diseased CSSTs (485 ± 10 g) were obtained from a CSST farm in Hubei Province, China, in July 2022, and were immediately transported to the laboratory for the isolation and identification of the infection-causing microbe. In the experiment, we selected healthy CSSTs that were lively, had intact body surfaces, and had no pathogenic bacteria isolated from their livers. Healthy CSSTs were (480 ± 30 g) purchased from the same CSST farm in Hubei Province, China. The number of healthy CSSTs was 726. On the farm, CSSTs live in an outside pond with an outdoor temperature of (30 ± 1 °C) and a water temperature of (25 ± 1 °C). Food is fed once a day in the morning. In the laboratory, healthy CSSTs were raised in aquariums at a room temperature of (30 ± 1 °C) and a water temperature of (25 ± 1 °C). All animal experiments were performed in accordance with the animal experiment ethics review committee of the Experimental Animal Center of the Yangtze River Fisheries Research Institute, Chinese Academy of Fishery Sciences (ID Number: YFI 2022-zhouyong-28).

### 2.2. Isolation of Pathogens

Euthanasia was performed in water containing 250 mg/L of ethyl 3-aminobenzoate methane sulfonate (MS-222) (Sigma, St. Louis, MO, USA). Then the animal was removed, placed on ice, and the diseased CSST was dissected after surface disinfection with 70% alcohol. The livers of diseased CSST were sampled in a biosafety cabinet (ESCO, Changi, Singapore) using a sterile inoculation loop. The liver samples were inoculated on agar plates consisting of brain heart infusion medium (BHI, Weymouth, MA, USA) and cultured upside down at 30 °C for 24 h. Single colonies were picked from the plates and re-cultured on BHI agar plates, and the plates were incubated under the same conditions to obtain purified single colonies. The single colonies were separately inoculated in 5 mL of BHI liquid medium and incubated at 30 °C and 200 rpm for 24 h. Glycerol stocks of the bacterial culture obtained from each purified colony were prepared in 1.5 mL sterile Eppendorf tubes, and stored in an ultra-low temperature freezer at −80 °C. The bacterial isolate was named as XC-1908.

### 2.3. Identification of the Pathogen

#### 2.3.1. Morphological Characterization

We analyzed the morphological characteristics of isolate XC-1908 using a light microscope (Olympus, Tokyo, Japan). The colonies of isolate XC-1908 were resuspended in phosphate buffer solution (PBS, Procell, Wuhan, China); the mixture was smeared on glass slides, air-dried, and subjected to Gram staining (Jiancheng, Nanjing, China) [14]. For further morphological analysis, the bacterium was fixed using 2.5% glutaraldehyde solution(HEAD, Beijing, China), dehydrated, dried [15], and observed using a scanning electron microscope (Hitachi, Tokyo, Japan).

#### 2.3.2. Biochemical Characterization of Isolate XC-1908

Isolate XC-1908 was cultured in BHI solid medium, and a single colony was inoculated in Biolog universal growth agar (BUG, Thinkfar, Wuhan, China) identification plates; the plates were incubated at 28 °C for 16–24 h, and a colony of suitable size was resuspended in IF-A inoculation fluid provided in the Biolog Bacterial Identification Kit (Biolog, Hayward, CA, USA). The turbidity of the solution was measured using the Biolog turbidimeter and adjusted (between 92% T and 98% T; as per Biolog). The inoculation fluid with the resuspended bacterial colony was added to a GEN III plate (100 μL per well) using a pipette, and the plate was incubated in the Biolog system, which recorded the absorbance of each well at different time-points, and identified the isolate.

#### 2.3.3. 16S rRNA Gene Sequencing

Bacteria cultured at BHI for 24 h were harvested by centrifugation at 4000 rpm for 2 min, and genomic DNA of isolate XC-1908 was extracted from the bacterial cell pellet using a Bacterial DNA Kit (Tiangen, Beijing, China) as per the manufacturer’s instructions. PCR to amplify the 16S rRNA gene of the isolate was performed using universal primers (27F: 5′-AGAGTTTGATCATGGCTCAG-3′, 1492R: 5′-TACGGTTACCTTGTTACGACTT-3′) [16] for the 16S rRNA gene. The total volume of the reaction mixture was 25 μL, including 12.5 μL of PCR mix, 1 μL each of 10 μmol/L upstream and downstream primers, 1 μL of template DNA, and 9.5 μL of double-distilled water. The PCR conditions were as follows: pre-denaturation at 94 °C for 3 min, followed by 35 cycles of denaturation at 94 °C for 30 s, annealing at 55 °C for 30 s, and denaturation at 72 °C for 45 s, 35 cycles, and final extension at 72 °C for 10 min. The amplicons were confirmed by 1% agarose gel electrophoresis and sequenced. The sequencing results were placed on NCBI (National Center for Biotechnology Information) (https://www.ncbi.nlm.nih.gov, accessed on 1 March 2023) for sequence homology matching, and then the 16S rRNA gene sequences of *A. veronii* from different sources were taken to construct a phylogenetic tree by Neighbor-Joining (NJ) in MEGA7.0 (http://www.megasoftware.net/previousVersions.ph, accessed on 1 March 2023) for confidence testing with 1000 bootstrap analysis.

### 2.4. Biochemical Analysis of Serum Samples of CSSTs

A total of 3 mL of fresh blood was collected from the severed heads of three anesthetized naturally diseased CSSTs. A total of 3 mL of fresh blood was collected from the severed heads of the same three anesthetized healthy CSSTs and placed in separate Eppendorf tubes, which were well labeled. Repeat this experiment three times. The supernatant was obtained after being placed at 4 °C overnight and 4000 rpm for 10 min. The supernatant was transferred to a new Eppendorf tube [17], the levels of aspartate aminotransferase (AST), alanine aminotransferase (ALT) and alkaline phosphatase (ALP) activities, total protein (TP), albumin (ALB), and globulin (GLB) in the serum samples were measured using a fully automated biochemical analyzer (Sysmex, Kobe, Japan). Perform three replicate experiments.

### 2.5. Artificial Infection of CSSTs Using Strain XC-1908

Isolate XC-1908 was cultured for 24 h, and the concentration of the bacterium in the broth was determined using the plate counting method. Healthy CSSTs were randomly divided into six groups (one control group and five infection groups) with 30 turtles/group. For artificial infection, five groups of CSSTs were formed based on the concentration of the bacterium used for infection (1 × 10^4^, 1 × 10^5^, 1 × 10^6^, 1 × 10^7^, and 1 × 10^8^ colony-forming unit CFU/g of CSST). The control group was injected with the same volume of PBS. After anesthesia with MS-222, the control group was injected with 0.5 mL of sterile PBS in the peritoneal cavity of the CSSTs, and the test group was injected with 0.5 mL of different concentrations of bacteria in the peritoneal cavity of the CSSTs, respectively. After infection, they were observed continuously for 10 days at room temperature (30 ± 1 °C) in an aquarium with a water temperature of (25 ± 1 °C) and under normal air and light conditions. The number of deaths and symptoms were recorded. The median lethal (LD_50_) dose of strain XC-1908 to CSST was calculated using the Reed Muench method [18]. Three replicate experiments were performed.

### 2.6. Histopathological Analysis

Three healthy and three naturally diseased CSSTs were dissected after anesthesia. Small pieces of liver, spleen, ovaries, kidney, and intestine were taken separately and fixed in 4% paraformaldehyde for 24 h. The fixed samples were washed in running water for 12 h and then dehydrated with an ethanol gradient. The dehydrated samples were embedded in paraffin and cut into 5-μm-thick sections. After unfolding and drying on slides, the sections were stained with hematoxylin-eosin (HE) and observed using a light microscope (Olympus, Tokyo, Japan) [19].

### 2.7. Antibiotic Susceptibility Testing

Isolate XC-1908 was cultured for 24 h, and the concentration of the inoculum was adjusted to 1 × 10^8^ CFU/g using sterile PBS. One hundred microliters of the solution was spread on BHI agar plates, and the plates were undisturbed for 10 min until the solution on the surface of the medium was completely absorbed. The drug-sensitive paper sheets (Hangwei, Hangzhou, China) were placed on the plates, and the plates were incubated in a thermostatic incubator (CIMO, Shanghai, Chain) at 28 °C for 24 h. The size of the inhibition zone was measured, and sensitivity was determined according to the instructions of the drug-sensitive paper sheets.

## 3. Results

### 3.1. Clinical Symptoms of Diseased CSSTs

The diseased CSSTs exhibited the following symptoms: floated on the surface of the pond, moved slowly, and exhibited ulcerated skin on their abdomen. Histological analysis revealed thin and sticky blood; large, red, swollen, and bleeding pharyngeal; and enlarged liver, spleen, and kidneys. Moreover, the intestines exhibited bleeding without food (Figure 1).

### 3.2. Morphological Characterization of the Isolated Bacterium

The colonies of isolate XC-1908 were yellow in color, had a peculiar odor, and their surface was moist and smooth. They exhibited hemolysis on blood agar. Moreover, the bacterium was gram-negative and rod-shaped, without any budding cells or pods (Figure 2C). Scanning electron microscopy revealed that the bacterium was arc-shaped and approximately 2 μm in length (Figure 2D).

### 3.3. Pathogenicity of Isolate XC-1908

We observed that all concentrations of isolate XC-1908 caused mortalities. The highest mortality rate was observed in the 1 × 10^8^ CFU/g group, with 94% mortality. The control group exhibited no mortality during the experimental period. The LD_50_ of isolate XC-1908 after intraperitoneal injection in CSSTs was 4.17 × 10^5^ CFU/g. The clinical signs of CSSTs after injection of strain XC-1908 were consistent with those of naturally infected CSSTs. The mortality and survival curves of infected CSSTs are shown in Figure 3.

### 3.4. 16S rRNA Gene Sequencing Analysis

As per gel electrophoresis, the length of the 16S rRNA gene amplicon of isolate XC-1908 was 1407 bp. Moreover, 16S rRNA gene sequence analysis revealed that isolate XC-1908 exhibited more than 99% homology with isolates *A. veronii* (MN220557.1) and *A. veronii* (MN752428.1) as per the NCBI database. Moreover, phylogenetic analysis revealed that isolate XC-1908 was located on the same branch as *A. veronii* [20] (Figure 4).

### 3.5. Biochemical Characterization of Isolate XC-1908

Isolate XC-1908 was identified as *A. veronii* using the Biolog fully automated microbial identification system. The biochemical characteristics of the isolate are listed in Table 1.

### 3.6. Histopathological Changes in Diseased CSSTs

The liver, spleen, kidney, ovary, and intestine of the diseased CSSTs exhibited significant histopathological changes compared with those of the healthy CSSTs. Healthy turtles had intact cell structure in the liver, kidneys, intestines, spleen, and ovaries; color morphology of the tissues was normal, and no congestion was observed. Diseased CSSTs exhibited varying degrees of lesions in the tissues. The kidneys exhibited a large number of erythrocytes in the renal interstitium, and the renal glomerulus was edematous with erythrocyte infiltration (Figure 5B). High erythrocyte infiltration was observed in the infected spleen parenchyma; the red pulp was enlarged, and the necrotic splenocytes exhibited marginated nuclei (Figure 5D). The intestinal villi were disorganized, and the mucosal epithelial cells of the villi shed. Hemorrhaging was observed in the intestinal submucosa, muscle, and serosa (Figure 5F). The liver exhibited a large number of erythrocytes and inflammatory cells and high hemosiderin levels, and the number of vacuoles increased (Figure 5H). Infected oocytes exhibited red and round particles and increase in the number of vacuoles. Blood vessels around the oocytes were dilated and congested (Figure 5J). It was found that the pathological changes in various tissues of the CSSTs after the injection of the XC-1908 bacterium were the same as those of the natural-onset CSSTs.

### 3.7. Biochemical Analysis of Serum Samples

Biochemical analysis of the serum samples of affected CSSTs revealed that the levels of total protein, albumin, and globulin were 25.6 g/L, 7.6 g/L, and 18 g/L, respectively, which were significantly lower than those of healthy CSSTs. The levels of aspartate aminotransferase, alanine aminotransferase, and alkaline phosphatase were 594 g/L, 559 g/L, and 23.5 g/L, respectively, in the affected CSSTs, which were significantly higher than those of the healthy CSSTs (Figure 6).

### 3.8. Antibiotic Susceptibility Analysis

It was observed that the isolated strain *A. veronii* XC-1908 was sensitive to ceftriaxone, doxycycline, flupenthixol, cefradine, and gentamicin; moderately sensitive to enrofloxacin, neomycin, ampicillin, meldimycin, and polymyxin; and resistant to sulfonamide, carbenicillin, benzocillin, clindamycin, erythromycin, and streptomycin (Table 2).

## 4. Discussion

Recently, *A. veronii* infections have increased in several varieties of fish [21], such as *Lateolabrax maculatus* [22], *Loach misgurnus anguillicaudatus* [23], and *Poecilia reticulata* [24]. Different species of fish infected by *A. veronii* exhibit different symptoms, but the main symptoms were skin ulcers, organ bleeding, and severe ascites. When *Cilurus asotus*, *Gadus*, *Cyprinus carpio*, and *Caridina cantonensis* were infected with *A. veronii*, the main symptoms are scale shedding and ulceration in severe cases [25]. *A. veronii* can also infect amphibians and can cause skin decay in giant salamanders [26]. In the present study, the CSSTs infected with *A. veronii* exhibited ulceration of the abdomin, extensive redness, and bleeding of the pharyngeal, enlarged liver, enlarged kidneys, and bleeding of the intestine without food. Infection of CSSTs by *A. veronii* has been reported, and virulence genes, antibiotic susceptibility, and PCR have been described in diseased CSSTs [27]. In addition, we performed morphological analysis and bacterial physiological and biochemical identification of this bacterium. The isolated XC-1908 was Identified as *A. veronii*.

The LD_50_ of *A. veronii* varies for different aquatic animals [28]. The LD_50_ for longsnout catfish and largemouth bass was 3.47 × 10^4^ CFU/g [29] and 3.72 × 10^4^ CFU/g [30], respectively. In the present study, the LD_50_ of *A. veronii* XC-1908 for CSST was 4.17 × 10^5^ CFU/g, which is higher than that for longsnout catfish and largemouth bass. *A. veronii* XC-1908 was peritoneally injected into CSSTs, and the clinical symptoms were consistent with those observed in case of natural infection.

Pathological diagnosis is an important clinical diagnostic tool. It is also used for the pathological analysis of aquatic animals [31]. In *A. veronii*-infected crucian carp, cell necrosis in several tissues and organs, internal congestion of blood vessels, and the infiltration of inflammatory cells in the kidneys was observed [21]. After being infected by *A. veronii*, the liver samples of Nile tilapia [32] exhibited histopathological changes with accumulation of iron-containing heme, and the spleen of *Danio rerio* exhibited infiltration of erythrocytes [33]. In the present study, histopathological analysis elucidated that the intestinal villi were shed, large number of erythrocytes and inflammatory cells infiltrated the spleen and kidney interstitium, and the number of inflammatory cells and the levels of hemosiderin increased in the liver. Similar histopathological features were observed in *Cyprinus carpio*, *Tilapia nilotica*, and *Danio rerio* infected by *A. veronii* in previous studies.

The serum samples of animals are important indicators of their physiological and biochemical condition [34]. In the serum biochemical subanalysis of *Pseudosciaena crocea* infected by *Vibrio harveyi* [35], the levels of TP, ALB, and GLB decreased, whereas those of AST and AKP increased. In the present study, TP, ALB, and GLB levels decreased (*p* < 0.05), suggesting that the CSSTs were infected and their immune system was compromised. Moreover, the levels of AST, AKP, and ALT were remarkably elevated, indicating that the liver and heart muscle of the diseased CSSTs were inflamed. These results are in accordance with the results of the biochemical analyses of the serum samples of diseased *Pseudosciaena crocea* [35].

The K-B paper diffusion method for testing drug sensitivity is a useful technique in the field of bacteriology and clinical pharmacology [36]. In reported reports of *A. veronii* infection in CSSTs, the results of drug susceptibility tests indicated that *A. veronii* was sensitive to streptomycin. However, in the present study, the pathogenic bacterium *A. veronii* XC-1908 was sensitive to ceftriaxone, doxycycline, florfenicol, cefradine, and gentamicin; moderately sensitive to enrofloxacin, neomycin, ampicillin, methicillin, and polymyxin; and resistant to sulfonamide, carbenicillin, benzocillin, clindamycin, erythromycin, and streptomycin. The sensitivity results obtained in this study differ from those already reported for drug sensitivity [27]. This may be because resistance against antibiotics varies with the genotype of the isolate. According to the results of the drug sensitivity test in this experiment and the comparison of previous literature. We found that even the same pathogenic bacteria can have different drug resistance because of their different sources. Therefore, we suggest that for the prevention and control of bacterial diseases in CSST culture, daily monitoring of bacterial drug resistance should be done so that farmers have effective antibiotic species. In the event of a disease outbreak, the correct drugs should be used to control the disease in the first instance. In the process of breeding, it is important to do a good drug sensitivity test, targeted drug use, and keep good records when using drugs. We should try to use a combination of narrow-spectrum antibiotics, combination, and rotation of drugs.

## 5. Conclusions

In this study, bacterium *A. veronii* XC-1908 was isolated from infected CSSTs. The LD_50_ of the bacterium was 4.17 × 10^5^ CFU/g for CSSTs. It was highly pathogenic to the CSSTs and caused internal bleeding. The findings of this study will provide new insights into the diagnosis and treatment of infections caused by *A. veronii* in aquatic animals and will elucidate the pathogenesis of *A. veronii* infections in CSSTs.

## Figures and Tables

**Figure 1 microorganisms-11-01304-f001:**
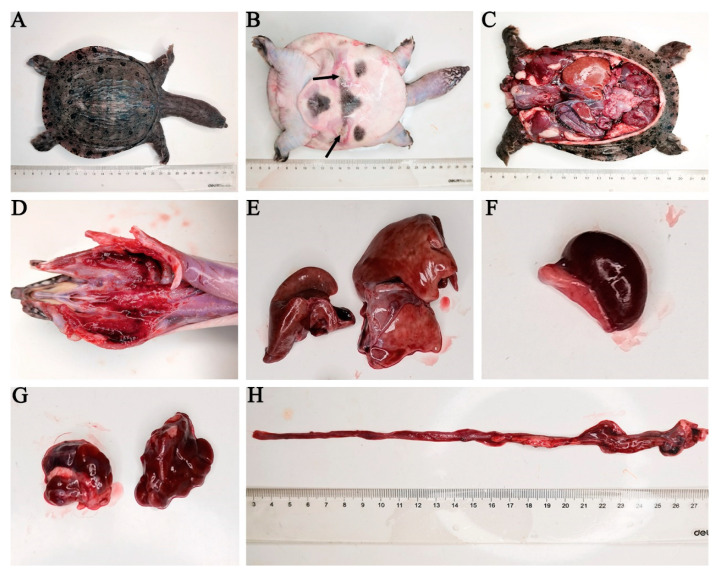
Clinical symptoms of a diseased CSST. (**A**) The back of a diseased CSST, (**B**) abdominal ulceration, (**C**) anatomical picture of a diseased CSST, (**D**) redness and bleeding in the gills, (**E**) enlarged liver, (**F**) enlarged spleen, (**G**) kidneys of a diseased CSST, (**H**) bleeding in the intestine. The black arrow indicates the ulceration point.

**Figure 2 microorganisms-11-01304-f002:**
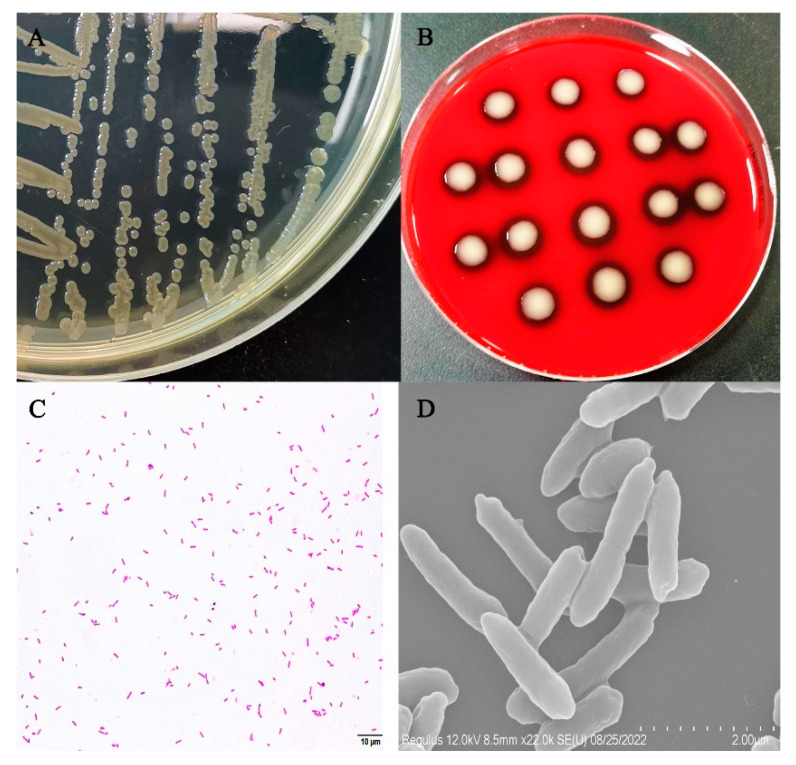
Morphological characteristics of isolate XC-1908. (**A**) Isolate XC-1908 cultured on BHI agar, (**B**) isolate XC-1908 cultured on blood agar (HopeBio, Qingdao, China), (**C**) image of Gram-stained isolate XC-1908 (scale bar: 10 μm), (**D**) scanning electron micrograph of isolate XC-1908 strain (scale bar: 2 μm).

**Figure 3 microorganisms-11-01304-f003:**
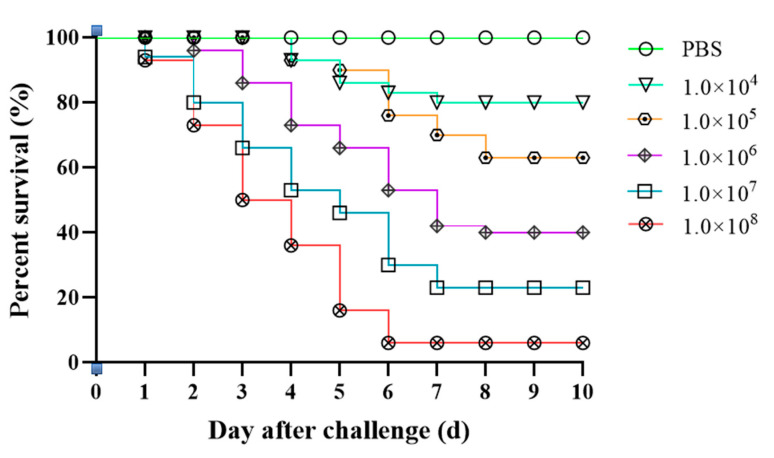
Regression infection analysis of CSSTs using isolate XC-1908. Survival rate of CSSTs after injection with different concentrations of inoculum (isolate XC-1908). The control group was intraperitoneally injected with 0.5 mL of sterile PBS, and the test group was intraperitoneally injected intraperitoneally with 0.5 mL of different concentrations of bacterial culture (1 × 10^4^, 1 × 10^5^, 1 × 10^6^, 1 × 10^7^, 1 × 10^8^ colony-forming unit CFU/g of CSST); both groups were monitored for 10 days.

**Figure 4 microorganisms-11-01304-f004:**
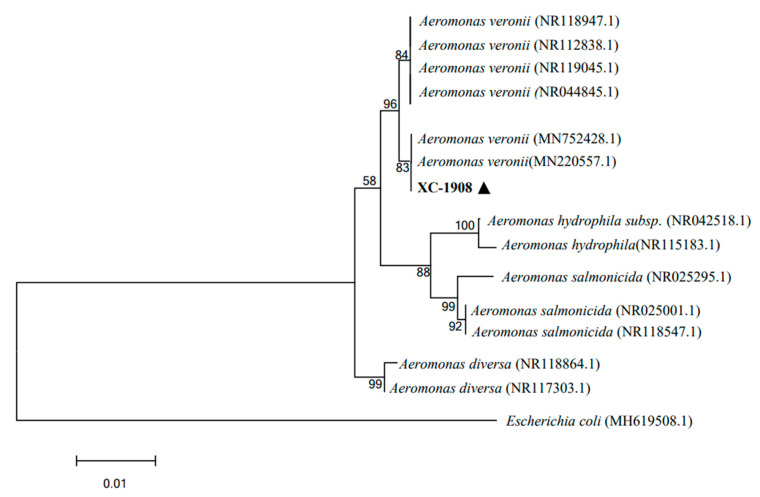
Phylogenetic analysis of 16S rRNA gene sequence of isolate XC-1908. The phylogenetic tree was constructed using the neighbor-joining method in MEGA 7.0. 16S rRNA gene sequence of isolate XC-1908 was aligned with those of nine members of the genus *Aeromonas*; the sequences were obtained from the NCBI database. The number at each branch indicates the percentage of bootstrap values for 1000 replicates. The scale bar indicates the number of substitutions per site. The black triangle indicates the isolated pathogenic bacteria.

**Figure 5 microorganisms-11-01304-f005:**
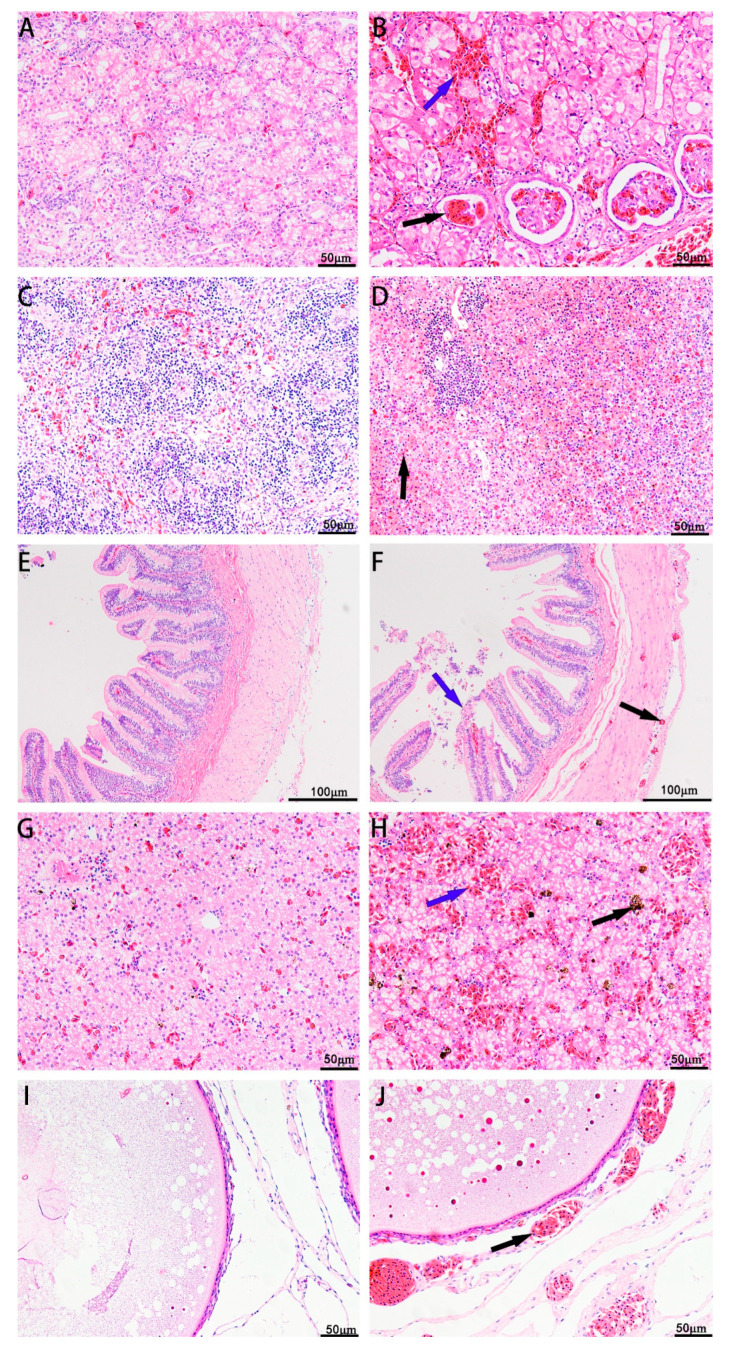
Histopathological observations from healthy and diseased CSSTs. (**A**) Kidneys of a healthy CSST, (**B**) kidneys of a diseased CSST infiltrated with a large number of inflammatory cells, glomerulus atrophy; blue arrows indicate erythrocytes, and black arrow indicated renal glomerulus, (**C**) spleen of a healthy CSST, (**D**) diseased CSST spleen with a large number of red blood cells and inflammatory cells; black arrow indicates erythrocytes, (**E**) intestine of a healthy CSST, (**F**) intestinal villi shedding in a diseased CSST; blue arrow indicates epithelial cells, and black arrow indicates bleeding point, (**G**) liver of a healthy CSST, (**H**) liver of diseased CSST with high hemosiderin levels; blue arrow indicates erythrocytes, and black arrow indicates hemosiderin, (**I**) eggs of a healthy CSST, (**J**) increase in number of vacuoles in egg cells of a diseased CSST; blood vessels dilated and congested; black arrow indicates blood vessels. Scar bar = 50 μm for (**A**–**D**,**G**–**I**); for (**E**,**F**), scale bar = 100 μm.

**Figure 6 microorganisms-11-01304-f006:**
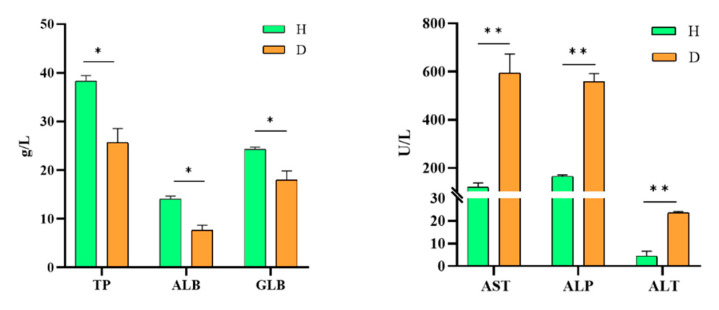
Biochemical analyses of serum samples of isolate XC-1908-infected CSSTs. TP: total protein, ALB: albumin, GLB: globulin, AST: aspartate aminotransferase, AKP: alkaline phosphatase, ALT: alanine aminotransferase, H: healthy CSST, D: diseased CSST. (* *p* > 0.05,** *p* < 0.01).

**Table 1 microorganisms-11-01304-t001:** Biochemical characteristics of isolate XC-1908 as per Biolog.

Reaction	Result	Reaction	Result
A1 Negative Control	N	E1 Gelatin	B
A2 Dextrin	P	E2 Glycyl-L-Proline	B
A3 D-Maltose	B	E3 L-Alanine	B
A4 D-Trehalose	P	E4 L-Arginine	B
A5 D-Cellobiose	B	E5 L-Aspartic Acid	B
A6 Gentiobiose	B	E6 L-Glutamic Acid	B
A7 Sucrose	B	E7 L-Histidine	B
A8 D-Turanose	B	E8 L-Pyroglutamic Acid	B
A9 Stachyose	B	E9 L-Serine	B
A10 Positive Control	B	E10 Lincomycin	N
A11 Acidic PH PH6	B	E11 Guanidine HCl	B
A12 Acidic PH PH5	N	E12 Niaproof 4	B
B1 D-Raffinose	B	F1 Pectin	B
B2 α-D-Lactose	B	F2 D-Galacturonic Acid	B
B3 D-Melibiose	B	F3 L L-Galactonic Acid Lactone	B
B4 *β*-Methyl-D-Glucoside	B	F4 D-Gluconic Acid	B
B5 D-Salicin	B	F5 D-Glucuronic Acid	B
B6 N-Acetyl-D-Glucosamine	B	F6 Glucuronamide	N
B7 N-Acetyl-*β*-D-Mannosamine	B	F7 Mucic Acid	B
B8 N-Acetyl-D-Galactosamine	N	F8 Quinic Acid	B
B9 N-Acetyl Neuraminic Acid	B	F9 D-Saccharic Acid	B
B10 1% NaCl	B	F10 Vancomycin	B
B11 4% NaCl	N	F11 Tetrazolium Violet	B
B12 8% NaCl	N	F12 Tetrazolium Blue	B
C1 α-D-Glucose	B	G1 P-Hydroxy-Phenylacetic Acid	N
C2 D-Mannose	B	G2 Methyl Pyruvate	B
C3 D-Fructose	B	G3 D-Lactic Acid Methyl Ester	B
C4 D-Galactose	B	G4 L-Lactic Acid	B
C5 3-Methyl Glucose	N	G5 Citric Acid	B
C6 D-Fucose	N	G6 6α-Keto-Glutaric Acid	B
C7 L-Fucose	N	G7 D-Malic Acid	N
C8 L-Rhamnose	N	G8 L-Malic Acid	B
C9 Inosine	N	G9 Bromo-Succinic Acid	B
C10 1% Sodium Lactate	B	G10 Nalidixic Acid	B
C11 Fusidic Acida	N	G11 Lithium Chloride	B
C12 D-Serine	B	G12 Potassium Tellurite	N
D1 D-Sorbitol	B	H1 Tween 40	B
D2 D-Mannitol	B	H2 *γ*-Amino-Butyric Acid	B
D3 D-Arabitol	N	H3 *α*-Hydroxy-Butyric Acid	B
D4 myo-lnositol	B	H4 β-Hydroxy-D,L-Butyric Acid	N
D5 Glycerol	B	H5 α-Keto-Butyric Acid	B
D6 D-Glucose-6-PO4	B	H6 Acetoacetic Acid	N
D7 D-Fructose-6-PO4	B	H7 Propionic Acid	N
D8 D-Aspartic Acid	N	H8 Acetic Acid	B
D9 D-Serine	B	H9 Formic Acid	N
D10 Troleandomycin	B	H10 Aztreonam	N
D11 Rifamycin SV	B	H11 Sodium Butyrate	N
D12 Minocycline	N	H12 Sodium Bromate	N

Notes: “P” = Pos; “N” = Neg; “B” = Borderline.

**Table 2 microorganisms-11-01304-t002:** Drug sensitivity analysis of isolate XC-1908.

Drug	Concentration	Inhibition Zone (mm)	Sensitivity
Ceftriaxone	30 μg	34	S
Enrofloxacin	10 μg	13	I
Doxycycline	30 μg	18	S
Sulfanilamide	300 IU	8.5	R
Gentamicin	10 μg	16	S
Neomycin	30 μg	17	I
Fluphenazole	30 μg	32	S
Carbenicillin	100 μg	6	R
Cefradine	30 μg	20	S
Erythromycin	15 μg	7	R
Benazicillin	1 μg	6	R
Ampicillin	10 μg	11	I
Meldimycin	30 μg	15	I
Streptomycin	10 μg	7	R
Clindamycin	2 μg	6	R
Polymyxin	300 IU	12	I

Notes: S, susceptible; I, intermediate; R, resistant.

## Data Availability

The data presented in this study are available on request from the corresponding author.

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
