# Peer review of "Isolation, Identification, and Characterization of Aeromonas veronii from Chinese Soft-Shelled Turtle (Trionyx sinensis)"

_microorganisms, 2023, doi:10.3390/microorganisms11051304_

Round 1

Reviewer 1 Report

Dear managing editor,

I am very glad to receive your email inviting me to review manuscript microorganisms-2322151 submitted to Microorganisms. The topic is ‘Isolation, Identification, and Characterization of Aeromonas veronii from Chinese soft-shelled turtle (Trionyx sinensis)’. In the manuscriptXiaowei Hu et al. identified and morphologically analyzed the strain isolated from the Chinese soft-shelled turtle and performed drug sensitivity tests to elucidate the drug susceptibility of this bacterium. Helpful in the control of diseases of the Chinese soft-shelled turtle.

These results are interesting, and the study was well-constructed and the conclusions are valid, but when I read this manuscript thorough carefully, I also have found some errors in this manuscript, so I give my recommendation for minor revision.

All in all, I think this manuscript could be acceptable for the publication in the journal of animals after the minor revisions.

1.       Line 188-192:In the XC-1908 pathogenicity test, please indicate whether the clinical symptoms of the Chinese soft-shelled turtle were consistent with those of the natural onset of disease after the bacterial injection.

2.       Line 150: I don’t think paraformaldehyde is what was used – believe it should be formaldehyde or formalin.

3.       Take a closer look at the chart. In Figure 1, the photo of the Chinese soft-shelled turtle abdomen has black shear heads. Please point out what the black shear heads refer to.

4.       Refer to the journal's reference style for the exact formatting of these documents, as well as species names in the reference list should be italicized.. For example Pelodiscus sinensis and Trionyx sinensis to italics in references 3, 6 and 7.

5.       Line 100 “BUG” source? Line 121 “NCBI” website?

6.       Line 183: Figure 2 in the notes is bolded.

7.       Line 162: Thermostatic incubator should add to the production company. Blood agar is also consistent.

8.       It is suggested to change the positions of Figure 2C and Figure 2D

9.       Please supplement the Latin names of all the first species in the discussion and use the correct format (italic). E.g “catfish” “cod” “carp” “tiger crayfish” “zebrafish”etc.

10.    Line 70: The details of temporary conditions and diet of “Healthy CSSTs”?

11.    Line 125: Add a separator for numbers over 1,000. Check all numbers including those in the tables/figures.

Author Response

For detailed answers, please see the attachment

Reviewer 2 Report

The manuscript microorganisms-2322151 reported the experimental data on the isolation, identification and characterization of an Aeromonas veronii strain from Chinese soft-shelled turtle. The experimental results showed the morphological and biochemical characteristics of the bacteria, and the pathological section of the diseased Chinese soft-shelled turtle was observed. At the same time, the drug resistance experiment of this bacteria was carried out to screen out effective antibiotics. These results were consistent with Aeromonas veronii isolated from other animal models.

This topic is very important. The possible diseases in the breeding process of Chinese soft-shelled turtle were further supplemented, which increased the knowledge in this field. The manuscript is well organized. I think this manuscript is suitable for publication on ' microorganisms ', and I suggest that it be accepted with minor modifications. Some minor issues need to be addressed prior to accept this manuscript for publication:

1. Line 30: ‘A. veronii should be written as Aeromonas veronii

2. Line 68-70: The healthy and diseased Chinese soft-shelled turtles were obtained from the same farm or different farms. If they were obtained from different farms, whether the same pathogenic bacteria were isolated and identified from healthy Chinese soft-shelled turtles. Or how to ensure that healthy turtles are completely healthy.

3. Line 93: PBS should be added to the production company. Line 95: Glutaraldehyde solution is also consistent.

4. Line 100: BUG should be written as ‘Biolog universal growth agar (BUG)’

5. Line 141: The control group was injected with PBS, but the concentration of bacteria was not mentioned in the previous dilution with PBS, please add the note.

6. Line 146: Chinese soft-shelled turtle is sensitive to changes in ambient temperature. What is the basis for the selection of experimental temperature and whether it has adverse effects on Chinese soft-shelled turtle.

7. Line 191: Calculation method of supplementary LD50.

8. The white arrow in Figure 5 is not obvious in the picture, please change other colors. And whether the results of pathological sections were compared with those of infected Chinese turtles after injection, and whether the symptoms were consistent.

9. Line 287: Which is lower than that for other aquatic animals. Does this mean that LD50 is lower than other studies or less toxic than bacteria in other studies? Please modify.

10. Authors are requested to check the format of all references, and species names should be italicized.

Author Response

(The authors gave the same response as above.)

Reviewer 3 Report

The authors must add a more detailed protocol for enzyme and protein identification from blood samples (section 2.4), deposit 16S rRNA gene sequence into the NCBI, and correct the name of the bacterium in the title into italic.

Author Response

(The authors gave the same response as above.)

Reviewer 4 Report

The manuscript describes quite precisely (some information missing) techniques and findings of the isolation, identification and characterization of A. veronii in a case of infectious disease in a soft-shell farm. The infection is quite common in aquatic environments, and well known for trionyx as well, however the description of symptoms and lesions of the infection is interesting. The treatment of animals is not always ethically acceptable, the authors are asked to provide the reasons for their decisions. Authors announce “

This study provides control strategies to prevent outbreaks of A. veronii infection in CSSTs” but this discussion is missing.

References are updated, but unexpectedly lacking a comparable available citation. Tables and figures are appropriate.

38-44: For the western reader it would be useful to indicate the reasons for the farming of these turtles, evidently so developed as to derive "huge economic losses" in cases of mass mortality

2.1 section: please, report some information about farming conditions and housing of experimental turtles

68 and 70: Please, specify the number of diseased and healthy animals

128: were healthy turtles sacrificed to obtain a blood sample for biochemical panel? Blood analysis is a common clinical practice, it is not necessary (and not acceptable) to sacrifice animals to obtain the requested blood sample (it can be very small if vetscan is used, with <0.1 mL requested to run a full comprehensive panel). Again, please indicate animals' number

145-146: again, please specify housing conditions

169 and 274: Trionyx turtles have pharyngeal villiform papilla that act as respiratory (and presumably salt uptaking) organs, but not real gills, so please adjust definition in order to not confuse reader and for more scientific precision (please refer to Yoshie S, Yokosuka H, Kaneko T, Fujita T. The existence of Na+/K+-ATPase-immunoreactive cells in the pharyngeal villiform-papilla epithelium of the soft-shelled turtle, Trionyx sinensis japonicus. Arch Histol Cytol. 2000 Jul;63(3):285-90. doi: 10.1679/aohc.63.285, Yokosuka H, Ishiyama M, Yoshie S, Fujita T. Villiform processes in the pharynx of the soft-shelled turtle, Trionyx sinensis japonicus, functioning as a respiratory and presumably salt uptaking organ in the water. Arch Histol Cytol. 2000 May;63(2):181-92. doi: 10.1679/aohc.63.181; Yokosuka H, Murakami T, Ishiyama M, Yoshie S, Fujita T. The vascular supply of the villiform processes in the pharynx of the soft-shelled turtle, Trionyx sinensis japonicus. A scanning electron microscopic study of corrosion casts. Arch Histol Cytol. 2000 May;63(2):193-8. doi: 10.1679/aohc.63.193)

149 and 219: was it necessary to sacrifice healthy animals to compare normal and diseased histology? A skilled pathologist does not require this procedure, and after all the findings were “Healthy turtles had intact cell structure in the liver, kidneys, intestines, spleen, and ovaries; color morphology of the tissues was normal, and no congestion was observed”…

273-278: A. veronii is a well-known pathogen for soft shell turtles, and has been characterized for virulence genes, antibiotic sensitivity and PCR in diseased T. sinensis (see Ye Y, Jiang Y, Fan T, Jiang Q, Cheng Y, Lu J, Lin L. Resistance characterization, virulence factors, and ERIC-PCR fingerprinting of Aeromonas veronii strains isolated from diseased Trionyx sinensis.Foodborne Pathog Dis. 2012 Nov;9(11):1053-5. doi: 10.1089/fpd .2012.1181)

282 and 286-296: reptiles and fishes have very different physiologies, the correlation is not appropriate, as the only similarity is the aquatic environment

298-300 and 304-305: a fish species and a different pathogen is not a correct correlation. Please, refer to chelonians' diseases

307-315: references are available for sensitivity test of A. veronii isolates from T. sinensis (see comment to 273-278), please refer to compatible data

316-319: spreading of antibiotic resistance is a global problem, please spend a few more words on the argument and on the announced “control strategies to prevent outbreaks of A. veronii infection in CSSTs”

Author Response

(The authors gave the same response as above.)
